# Genome-wide identification of directed gene networks using large-scale population genomics data

René Luijk[1], Koen F. Dekkers[1], Maarten van Iterson[1], Wibowo Arindrarto [2], Annique Claringbould[3], Paul Hop[1], BIOS Consortium[#], Dorret I. Boomsma[4], Cornelia M. van Duijn[5], Marleen M.J. van Greevenbroek[6,7], Jan H. Veldink[8], Cisca Wijmenga[3], Lude Franke [3], Peter A.C. 't Hoen [9], Rick Jansen [10], Joyce van Meurs[11], Hailiang Mei[2], P. Eline Slagboom[1], Bastiaan T. Heijmans [1] & Erik W. van Zwet[12]

Identification of causal drivers behind regulatory gene networks is crucial in understanding gene function. Here, we develop a method for the large-scale inference of gene–gene interactions in observational population genomics data that are both directed (using local genetic instruments as causal anchors, akin to Mendelian Randomization) and specific (by controlling for linkage disequilibrium and pleiotropy). Analysis of genotype and whole-blood RNA-sequencing data from 3072 individuals identified 49 genes as drivers of downstream transcriptional changes (Wald $P < 7 \times 10^{-10}$), among which transcription factors were over-represented (Fisher's $P = 3.3 \times 10^{-7}$). Our analysis suggests new gene functions and targets, including for *SENP7* (zinc-finger genes involved in retroviral repression) and *BCL2A1* (target genes possibly involved in auditory dysfunction). Our work highlights the utility of population genomics data in deriving directed gene expression networks. A resource of *trans*-effects for all 6600 genes with a genetic instrument can be explored individually using a web-based browser.

[1] Molecular Epidemiology Section, Department of Medical Statistics and Bioinformatics, Leiden University Medical Center, Leiden, Zuid-Holland 2333 ZC, The Netherlands. [2] Sequence Analysis Support Core, Leiden University Medical Center, Leiden, Zuid-Holland 2333 ZC, The Netherlands. [3] Department of Genetics, University of Groningen, University Medical Centre Groningen, Groningen 9713 AV, The Netherlands. [4] Department of Biological Psychology, VU University Amsterdam, Neuroscience Campus Amsterdam, Amsterdam 1081 TB, The Netherlands. [5] Genetic Epidemiology Unit, Department of Epidemiology, ErasmusMC, Rotterdam 3015 GE, The Netherlands. [6] Department of Internal Medicine, Maastricht University Medical Center, Maastricht 6211 LK, The Netherlands. [7] School for Cardiovascular Diseases (CARIM), Maastricht University Medical Center, Maastricht 6229 ER, The Netherlands. [8] Department of Neurology, Brain Center Rudolf Magnus, University Medical Center Utrecht, Utrecht 3584 CG, The Netherlands. [9] Department of Human Genetics, Leiden University Medical Center, Leiden, Zuid-Holland 2333 ZC, The Netherlands. [10] Department of Psychiatry, VU University Medical Center, Neuroscience Campus Amsterdam, Amsterdam 1081 HV, The Netherlands. [11] Department of Internal Medicine, ErasmusMC, Rotterdam 3015 CE, The Netherlands. [12] Medical Statistics Section, Department of Medical Statistics and Bioinformatics, Leiden University Medical Center, Leiden, Zuid-Holland 2333 ZC, The Netherlands. These authors jointly supervised this work: Bastiaan T. Heijmans, Erik W. van Zwet. [#]A full list of consortium members appears at the end of the paper. Correspondence and requests for materials should be addressed to B.T.H. (email: b.t.heijmans@lumc.nl) or to E.W.v.Z. (email: e.w.van_zwet@lumc.nl)

dentification of the causal drivers underlying regulatory gene networks may yield new insights into gene function[1,2], possibly leading to the disentanglement of disease mechanisms characterized by transcriptional dysregulation[3]. Gene networks are commonly based on the observed co-expression of genes. However, such networks show only undirected relationships between genes which makes it impossible to pinpoint the causal drivers behind these associations. Adding to this, confounding (e.g., due to demographic and clinical characteristics, technical factors, and batch effects[4,5]) induces spurious correlations between the expression of genes. Correcting for all confounders may prove difficult as some may be unknown[6]. Residual confounding then leads to very large, inter-connected co-expression networks that do not reflect true biological relationships.

To address these issues, we exploited recent developments in data analysis approaches that enable the inference of causal relationships through the assignment of directed gene–gene associations in population-based transcriptome data using genetic instruments[7–9] (GIs). Analogous to Mendelian Randomization[10,11] (MR), the use of genetics provides an anchor from where directed associations can be identified. Moreover, GIs are free from any non-genetic confounding. Related efforts have used similar methods to identify additional genes associated with different phenotypes, either using individual level data[7,8] or using publicly available eQTL and GWAS catalogues[9]. However, these efforts have not systematically taken linkage disequilibrium (LD) and pleiotropy (a genetic locus affecting multiple genes) into account. As both may lead to correlations between GIs, we aimed to improve upon these methods in order to minimize the influence of LD and pleiotropy, and would detect the actual driver genes. This possibly induces non-causal relations[12], precluding the identification of the specific causal gene involved when not accounted for LD and pleiotropy.

Here, we combine genotype and expression data of 3072 unrelated individuals from whole blood samples to establish a resource of directed gene networks using genetic variation as an instrument. We use local genetic variation in the population to capture the portion of expression level variation explained by nearby genetic variants (local genetic component) of gene expression levels, successfully identifying a predictive genetic instrument (GI) for the observed gene expression of 6600 protein-coding genes. These GIs are then tested for an association with potential target genes *in trans*. Applying a robust genome-wide approach that corrects for linkage disequilibrium and local pleiotropy by modelling nearby GIs as covariates, we identify 49 index genes each influencing up to 33 target genes (Bonferroni correction, Wald $P < 7 \times 10^{-10}$). Closer inspection of examples reveals that coherent biological processes underlie these associations, and we suggest new gene functions based on these newly identified target genes, e.g. for *SENP7* and *BCL2A1*. An interactive online browser allows researchers to look-up specific genes of interest (see URLs).

## Results

**Establishing directed associations in transcriptome data**. We aim to establish a resource of index genes that causally affect the expression of target genes *in trans* using large-scale observational RNA-sequencing data. However, causality cannot be inferred from the correlation between the observed expression measurements of genes, and therefore is traditionally addressed by experimental manipulation. Furthermore, both residual and unknown confounding can induce correlation between genes, possibly yielding to extensive correlation networks that are not driven by biology. To establish causal relations between genes, we assume a structural causal model[13] describing the relations

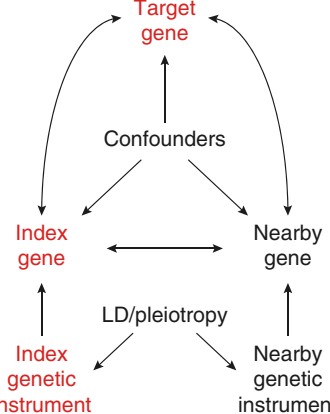

**Fig. 1** Diagram showing the presumed relations between each variable. A directed arrow indicates the possibility of a causal effect. For instance, the index genetic instrument represents nearby SNPs with a possible effect on the nearby gene (analogous to *cis*-eQTLs). A double arrow means the possibility of a causal effect in either direction. The index gene, for example, could have a causal effect on the target gene, or vice versa. We aim to assess the presence of a causal effect of the index gene on the target gene using genetic instruments (GIs) that are free of non-genetic confounding. To do this, we must block the back-door path from the index GI through the GIs of nearby genes to the target gene. This back-door path represents linkage disequilibrium and local pleiotropy and is precluded by correcting for the GIs of nearby genes. Correction for observed gene expression (either of the index gene or of nearby genes) does not block this back-door path, but instead possibly leads to a collider bias, falsely introducing a correlation between the index GI and the target gene

between genes and using their genetic components, the local genetic variants predicting their expression, as genetic instruments[10] (GIs). To be able to conclude the presence of a causal effect of the index gene on the target gene, the potential influence of linkage disequilibrium (LD) and pleiotropic effects have to be taken into account, as they may cause GIs of neighbouring genes to be correlated (Fig. 1). This is done by blocking the so-called back-door path[13] from the index GI through the genetic GIs of nearby genes to the target gene by correcting the association between the GI and target gene expression for these other GIs. Note that this path cannot be blocked by adjusting for the observed expression of the nearby genes, as this may introduce collider bias, resulting in spurious associations.

To assign directed relationships between the expression of genes and establish putative causality, the first step in our analysis approach was to identify a GI for the expression of each gene, reflecting the local genetic component. To this end, we used data on 3072 individuals with available genotype and gene expression data (Supplementary Data 1), measured in whole blood, where we focused on at least moderately expressed (see Methods) protein-coding genes (10,781 genes, Supplementary Fig. 1). Using the 1021 samples in the training set (see Methods), we obtained a GI consisting of at least 1 SNP for the expression of 8976 genes by applying lasso regression to nearby genetic variants while controlling for known (cohort, sex, age, white, and red blood cell counts) and unknown covariates[14] (see Methods). Adding distant genetic variants to the prediction model has been shown to add very little predictive power[7] and would have induced the risk of including long-range pleiotropic effects.

The strength of the GIs was evaluated using the 2051 samples in the test set (see Methods). Taking LD and local pleiotropy into account by including the GIs of neighbouring genes (<1 Mb, Fig. 1), we identified 6600 sufficiently strong GIs having at least partly specific predictive ability (Supplementary Fig. 2a) for the

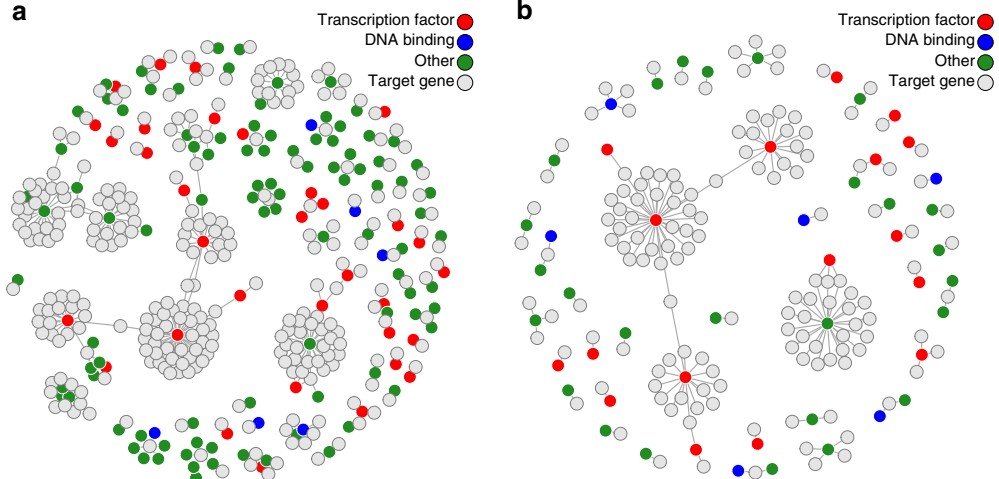

**Fig. 2** Gene networks showing the directed gene–gene association between genes. Panels show the associations when not taking LD and local pleiotropy into account (**a**) and when these are corrected for (**b**). Index genes identified as a transcription factor are indicated by red circles. Blue circles indicate index genes with DNA binding properties, but are not a known transcription factor[22]. Green circles indicate other index genes. Light grey circles indicate target genes. The uncorrected analysis shows 134 index genes (coloured circles) influencing 276 target genes, where several neighbouring index genes seemingly influencing the same target gene, which is reflective of a shared genetic component of those index genes. Specifically, 65 target genes are associated with multiple index genes which lie in close proximity to one another. The number of index genes drop sharply from 134 to 49 (2.7-fold decrease) when do taking LD and local pleiotropy into account. The number of target genes also drops, from 276 to 144 (1.9-fold decrease)

expression its corresponding index gene ($F$-statistic > 10, Supplementary Fig. 1, Supplementary Data 2). To evaluate the effects of these 6600 GIs on target gene expression, we tested for an association of each of 6600 GIs with all of 10,781 expressed, protein-coding genes *in trans* (>10 Mb, Supplementary Fig. 2b). To have maximum statistical power we used all 3072 samples, as opposed to only using the 2051 samples from the test set, as the results from both analyses showed very similar results (Supplementary Fig. 3). First, this analysis was done without accounting for LD and local pleiotropy (i.e., correcting for neighbouring LD, Fig. 1). This genome-wide analysis resulted in 401 directed associations between 134 index genes and 276 target genes after adjustment for multiple testing using the Bonferroni correction (Wald $P < 7 \times 10^{-10}$, Fig. 2, Supplementary Data 3). Among them were 134 index genes affecting the expression of 1 to 33 target genes *in trans* (3.2 genes on average, median of 1 gene), totalling 276 identified target genes. As expected, the resulting networks contained many instances where the same target gene was influenced by multiple neighbouring index genes, hindering the identification of the causal gene (65 such instances). Repeating the analysis for the 134 identified index genes, but corrected for LD and local pleiotropy by including the GIs of neighbouring genes (<1 Mb) resulted in the identification of specific directed effects for 49 index genes on 144 target genes, totalling 156 directed associations (Wald $P < 7 \times 10^{-10}$, Fig. 2), where the number of target genes affected by an index gene varied from 1 to 33 (Supplementary Data 8, 3.2 genes on average, median of 1 gene). The number of target genes associated with multiple neighbouring index genes drops from 65 to 2, underscoring the importance of correction for LD and pleiotropy. As this set of 156 directed associations is free from LD and local pleiotropy, and possibly reflect truly causal relations, we use these in further analyses.

**Validity and stability of the analyses**. To ensure the validity and stability of the analyses, we compared out methodology to earlier work and performed several checks regarding common challenges inherent to these analyses and the assumptions underlying them. First, we compared our approach to previously described approaches[7,8] by applying these to our data. Each approach

consists of a method to create GIs, and a model used to test for *trans*-effects. First, we used all methods to create GIs (lasso, elastic net, BLUP, and BSLMM), and investigated their predictive power of the index gene (see Methods). The methods that used feature selection (our method, lasso, and elastic net) showed similar predictive ability. Less predictive power was observed for methods using all nearby genetic variants (BLUP, BSLMM, Supplementary Fig. 4). Identifying *trans*-effects showed a lower number of *trans*-effects identified for the BLUP and BSLMM methods (Supplementary Fig. 5), possibly as a result of their less predictive GIs (Supplementary Fig. 4). In addition, as this *trans*-model does not take LD into account, a large number of target genes are associated with the GIs of many neighbouring index genes (Supplementary Fig. 5).

To investigate how well our proposed *trans*-model is able to control for LD, and to evaluate the statistical power of this model, we performed a simulation study investigating several scenarios (Supplementary Fig. 6, see Supplementary Methods for details on the simulation of the data). Overall, the simulations show high power to detect a true causal effect of the GI of the index gene on the target gene, where the correlation between GI and index gene, and between index gene and target gene contribute most to an increased power. The presence of correlated GIs of nearby genes plays a smaller role. Under the null hypothesis (i.e., when a neighbouring gene influences the target gene, and not the tested index gene, see blue and purple lines), the uncorrected analysis will indeed lead to false positives (indicated by higher power), while the corrected analysis will indeed lead to false positives in 5% of the tests performed, indicating LD is indeed corrected for. The simulation confirms that our approach is more specific in identifying the causal gene than its competitors.

By design, the GIs should be independent of most confounding factors, but confounding may still occur if genetic variants directly affect blood composition, leading to spurious associations. While we have already explicitly corrected for known white and red blood cell counts, we also evaluated the association of the 49 GIs with these cell counts, and found that none of the 49 GIs were significantly related to any observed cell counts (Supplementary Fig. 7a). In addition, all 156 directed associations

remained significant after further adjustment for nearby genetic variants (<1 Mb) reported to influence blood composition[15,16] (Supplementary Fig. 7b).

To combat any unknown residual confounding and possibly gain statistical power, we added five latent factors to our models, estimated from the observed expression data using cate[14] (see Methods). We re-tested the 156 identified associations without these factors to evaluate the model sensitivity, showing similar results with slightly attenuated test statistics (Supplementary Fig. 7c). This indicates that our analysis was not influenced by unknown confounding and confirmed the independence of GIs from non-genetic confounding, but did help in reducing the noise in the data, leading to increased statistical power.

Next, to validate the GIs of the 49 index genes, we compared the SNPs constituting the GIs of the 49 index genes associated with target gene expression with previous cis-eQTL mapping efforts. While similar sets of genes may be identified using a cis-eQTL approach, the utility of using multi-SNP GIs over single-SNP GIs (akin to cis-eQTLs) is shown in the increased predictive ability of multi-SNP GIs (Supplementary Fig. 7d), and thus in the number of predictive GIs. Only 4910 single-SNP GIs were predictive of its corresponding index gene (F-statistic > 10), compared to 6600 multi-SNP instrumental variables. Of the 49 index genes corresponding to the 49 GIs, 47 genes (96.1%) were previously identified as harbouring a cis-eQTL in large subset of the whole blood transcriptome data we analysed here (2116 overlapping samples), using an independent analysis strategy[17]. Almost all of the corresponding GIs (98%, 46 GIs) were strongly correlated with the corresponding eQTL SNPs ($R^2 > 0.8$). Similarly, 26 of the 49 index genes (53%) were also reported as having a cis-eQTL effect in a much smaller set of whole blood samples ($N_{GTEx} = 338$) part of GTEx[18], 23 of which also correlated strongly with the corresponding eQTL-SNPs ($R^2 > 0.8$). When considering all tissues in the GTEx project, we found 48 of 49 index genes were identified as harbouring a cis-eQTL in any of the 44 tissues measured.

Next, we compared our identified effects with trans-eQTLs identified earlier in whole-blood samples[19]. First, we found 97 target genes identified here (67%) overlapped with those found by Joehanes et al., 19 of which had their corresponding GI and lead SNP in close proximity (<1 Mb, Supplementary Fig. 8), suggesting that the effects are indeed mediated by the index gene assigned using our approach. Testing for a cis-eQTL of those SNPs identified by Joehanes et al. on the nearby index genes, we found all 19 index genes indeed had at least one nearby lead SNP that influenced its expression (Wald $P < 6 \times 10^{-4}$, Supplementary Data 4).

As a last check, we investigated potential mediation effects of each of the 49 GIs by observed index gene expression using the Sobel test[20] (Fig. 1). This method is based on the notion that the effect of a GI on target gene expression should diminish when correcting for the mediator observed index gene expression. However, small effect sizes and considerable noise in both mediator and outcome lead to low statistical power to detect mediated effects[21]. Nevertheless, we found 105 of 156 significant directed associations (67%) to show evidence for mediation (Bonferroni correction for 156 tests: Wald $P < 3.1 \times 10^{-4}$; Supplementary Data 5).

### Exploration of directed networks

To gain insight in the molecular function of 49 index genes affecting target gene expression, we used Gene Ontology (GO) to annotate our findings. The set of 49 index genes was overrepresented in the GO terms DNA Binding (Fisher's $P = 5 \times 10^{-8}$) and Nucleic Acid Binding (Fisher's $P = 2.8 \times 10^{-5}$, Supplementary Data 6), with 43.8% (21 genes) and 47.9% (23 genes) of genes overlapping with those gene sets, respectively. In line with this finding, we found a significant overrepresentation of transcription factors (17 genes; odds ratio = 5.7, Fisher's $P = 3.3 \times 10^{-7}$) using a manually curated database of transcription factors[22]. We note that such enrichments are expected a priori and hence indirectly validate our approach. Of interest, several target genes of two transcription factors overlapped with those identified in previous studies[23,24] (IKZF1: 27% of its target genes, 4 genes; PLAGL1: 15% of its target genes, 5 genes).

Finally, we explore the biological processes that are revealed by our analysis for several index genes that either are known transcription factors[22] or affect many genes in trans. While these results are limited to Bonferroni-significant directed associations (Wald $P < 7 \times 10^{-10}$, correcting for all possible combinations of the 6600 index genes and 10,781 target genes), researchers can interactively explore the whole resource using a dedicated browser (see URLs).

We identified 25 target genes to be affected in trans by sentrin/small ubiquitin-like modifier (SUMO)-specific proteases 7 (SENP7, Figs. 3 and 4, Supplementary Data 8), significantly expanding on the five previously suspected target genes resulting from an earlier expression QTL approach[25]. Increased SENP7 expression resulted in the upregulation of all but one of the target genes (96%). Remarkably, 23 of the 25 target genes affected by SENP7 are zinc finger protein (ZFP) genes located on chromosome 19. More specifically, 18 target genes are located in a 1.5 Mb ZFP cluster mapping to 19q13.43 (Fig. 3). ZFPs in this cluster are known transcriptional repressors, particularly involved in the repression of endogenous retroviruses[26]. Parallel to this, SENP7 has also been identified to promote chromatin relaxation for homologous recombination DNA repair, specifically through interaction with chromatin repressive KRAB-Association Protein (KAP1, also known as TRIM28). KAP1 had already been implicated in transcriptional repression, especially in epigenetic repression and retroviral silencing[27,28], although KAP1 had no predictive GI (F-statistic = 4.9). Therefore, it has been speculated SENP7 may also play a role in retroviral silencing[29]. Given the widespread effects of SENP7 on the transcription of chromosome 19-linked ZFPs involved in retroviral repression[26], it corroborates a role of SENP7 in the repression of retroviruses, specifically through regulation of this ZFP cluster. SENP7 is not a TF and does not bind DNA, but considering it is a SUMOylation enzyme, it possibly has its effect on the ZFP cluster through deSUMOylation of KAP1[30].

In our genome-wide analysis, we found that the transcription factor SP110 nuclear body protein (SP110) influences three zinc finger proteins (Figs. 3, 4). During viral infections in humans, SP110 has been shown to interact with the Remodelling and Spacing Factor 1 (RSF1) and Activating Transcription Factor 7 Interacting Protein (ATF7IP), suggesting it is involved in chromatin remodelling[31]. Interestingly, all three of the genes targeted by SP110 are also independently influenced by SENP7, although SP110 shows opposite effects (Supplementary Fig. 9), and are located in the same ZFP gene cluster on chromosome 19. This overlap of target genes supports the previous suggestion that SP110 is involved in the innate antiviral response[32], presumably through regulation of the same ZPF cluster regulated by SENP7.

The index gene with the most identified target gene effects in trans is Pleiomorphic Adenoma Gene-Like 1 (PLAGL1, also known as LOT1, ZAC). PLAGL1 is a transcription factor and affected 33 genes, 29 of which are positively associated with PLAGL1 expression (88%, Fig. 4). PLAGL1 is part of the imprinted HYMAI/ZAC1 locus, which has a crucial role in foetal development and metabolism[33]. This locus, and overexpression of PLAGL1 specifically, has been associated with transient neonatal diabetes mellitus[31,34] (TNDM) possibly by reducing

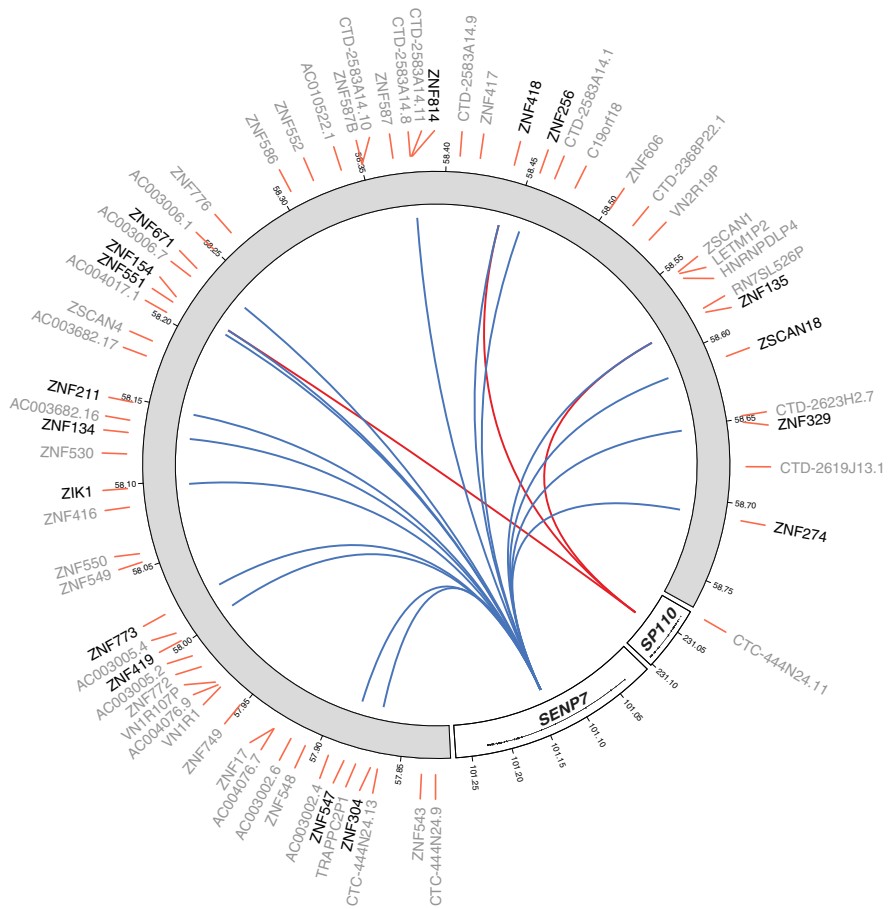

**Fig. 3** *SENP7* (chromosome 3) and *SP110* (chromosome 2) affect a zinc finger cluster located on chromosome 19. Many of these genes are involved in retroviral repression, among others. Blue lines indicate a positive association (upregulation), red lines indicate a negative association (downregulation). Colouring indicates consistent opposite effects of *SENP7* and *SP110* on their shared target genes

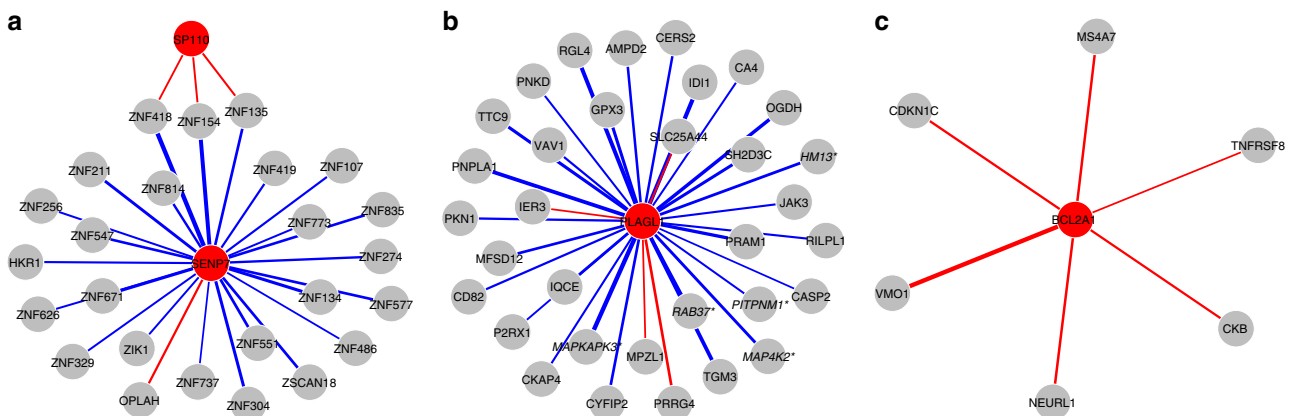

**Fig. 4** Identified target genes for different index genes. Panels show target genes for *SENP7* and *SP110* (**a**), *PLAGL1* (**b**), and *BCL2A1* (**c**). Starred and italic gene names indicate previously reported target genes[23, 24, 38, 80] (Supplementary Data 7). Blue and red lines indicate positive and negative associations, respectively; line thickness indicates strength of the association

insulin secretion[35]. *PLAGL1* is known to be a transcriptional regulator of PACAP-type I receptor[36] (*PAC1-R*). *PACAP*, in turn, is a regulator of insulin secretion[37]. In line with these findings, we found several target genes to be involved in metabolic processes. Most notably, we identified *MAPKAPK3* (MK3) and *MAP4K2* to be upregulated by *PLAGL1*, previously identified as *PLAGL1* targets[38], and both part of the mitogen-activated protein kinase (MAPK) pathway. This pathway has been observed to be upregulated in type II diabetic patients (reviewed in ref.[39]). In addition, inhibition of *MAPKAP2* and *MAPKAP3* in obese, insulin-resistant mice has been shown to result in improved metabolism[40], in line with the association between upregulation of *PLAGL1* and the development of TNDM. Furthermore, *PLAGL1* may be implicated in lipid metabolism and obesity through its effect on *IDI1*, *PNPLA1*, *JAK3*, and *RAB37* expression[41–44]. While not previously established as target genes, they are in line with the proposed role of *PLAGL1* in metabolism[33,45].

Increased expression of Bcl-related protein A1 (*BCL2A1*) downregulated all five identified target genes (Fig. 4). *BCL2A1* encodes a protein part of the B-cell lymphoma 2 (*BCL2*) family, an important family of apoptosis regulators. It has been implicated in the development of cancer, possibly through the inhibition of apoptosis (reviewed in ref.[46]). One target gene, *NEURL1*, is known to cause apoptosis[47], in line with its strong negative association with *BCL2A1* expression. Similarly, *CDKN1C* was also down-regulated by *BCL2A1*, and implicated in the promotion of cell death[48,49]. However, little is known about the strongest associated target gene, *VMO1* (Wald $P = 1.5 \times 10^{-8}$). It has been implicated in hearing, due to its highly abundant expression in the mouse inner ear[50], where *BCL2A1* may have a role in the development of hearing loss through apoptosis, since cell death is a known contributor to hearing loss in mice[51]. In line with its role in the inhibition of apoptosis, *BCL2A1* overexpression has a protective effect on inner ear mechanosensory hair cell death in mice[52]. Lastly, the target gene *CKB* has also been implicated in hearing impairment in mice[53] and Huntington's disease[54], further suggesting a role of *BCL2A1* in auditory dysfunction.

## Discussion

In this work, we report on an approach that uses population genomics data to generate a resource of directed gene networks. Our genome-wide analysis of whole-blood transcriptomes yields strong evidence for 49 index genes to specifically affect the expression of up to 33 target genes *in trans*. We suggest previously unknown functions of several index genes based on the identification of new target genes. Researchers can fully exploit the utility of the resource to look up *trans*-effects of a gene of interest using an interactive gene network browser (see URLs).

The identified directed associations provide improved mechanistic insight into gene function. Many of the 49 index genes affecting target gene expression are established transcription factors (TFs), or are known for having DNA-binding properties, an anticipated observation supporting the validity of our analysis. The identification of non-TFs will in part relate to the fact that the effect of an index gene may regulate the activity of TFs, for example by post-translational modification. This is illustrated by *SENP7* that we observed to concertedly affect the expression of zinc finger protein genes involved in the repression of retroviruses, likely by deSUMOylation of the transcription factor *KAP1*[30]. Other mechanistic insights that can be distilled from these results include the potential involvement of *BCL2A1* in auditory dysfunction, conceivably through the regulation of apoptosis.

While observational gene expression data can be used to construct gene co-expression networks[54], which is sometimes complemented with additional experimental information[38], such an approach lacks the ability to assign causal directions. Experimental approaches using CRISPR-cas9 coupled with single-cell technology[55–57] are in principle able to demonstrate putative causality at a large scale, but only in vitro, while the advantage of observational data is that it reflects in vivo situations. These experimental approaches currently rely on extensive processing of single-cell data that is associated with high technical variability[55], complicating the construction of specific gene-gene associations. In addition, off-target effects of CRISPR-cas9 cannot be excluded[58], potentially influencing the interpretation of these experiments. Finally, such efforts are currently limited in the number of genes tested[55–57], whereas we were able to perform a genome-wide analysis. Hence, experimental and population genomics approaches are complementary in identifying causal gene networks.

Traditional *trans*-eQTL studies aim to find specific genetic loci associated with distal changes in gene expression[19,59]. The limitation of this approach is that they are not designed to assign the specific causal gene responsible for the *trans*-effect because they do not control for LD and local pleiotropy (a genetic locus affecting multiple nearby genes). Hence, our approach enriches *trans*-eQTL approaches by specifying which index gene induces changes in target gene expression. However, it does not detect *trans*-effects independent of effects on local gene expression. In addition, identification of the causal path using a *trans*-eQTL approach is difficult to establish. Testing for mediation through local changes in expression[60,61] may be limited in statistical power, as these approaches are designed to only test the mediation effect of one lead SNP[60]. In addition, they too do not correct for pleiotropy or LD, possibly leading to several identified *cis*-genes mediating a *trans*-eQTL.

Related analysis methods were recently used to infer associations between gene expression and phenotypic outcomes (instead of gene expression as we did here). Two studies first constructed multi-marker GIs in relatively small sample sets to then apply these GIs in large datasets without gene expression data[7,8]. A different, summary-data-based Mendelian randomization (SMR) approach identifies genes associated with complex traits based on publicly available GWAS and eQTL catalogues[9]. However, neither of these approaches take LD or pleiotropic effects into account, led to many neighbouring, non-specific effects[7–9]. We show that correcting for LD and local pleiotropy will aid in the identification of the causal gene, as opposed to the identification of multiple, neighbouring genes, analogous to fine mapping in GWAS. Furthermore, the use of eQTL and GWAS catalogues are usually the result of genome-wide analyses, where only statistically significant variants are taken into account. Here, we use the full genetic landscape surrounding a gene, thereby maximizing the predictive ability of expression measurements by our GIs[7]. While we have used our genome-wide approach to identify directed gene networks, we note this method may also be used to annotate trait-associated variants with potential target genes, either by using individual level data[7,8], or by using SMR[9].

The analysis approach presented here relies on using GIs of expression of an index gene as a causal anchor, an approach akin to Mendelian randomization[10]. While GIs could provide directionality to bi-directional associations in observational data, genetic variation generally explains a relatively small proportion of the variation in expression (Supplementary Fig. 2a). The GIs for index gene expression identified here are no exception, significantly limiting statistical power of similar approaches[62,63]. Increased sample sizes and improvement on the prediction of index gene expression will help in identifying more target genes.

Our current analysis strategy aims for causal inference, obviating LD and pleiotropic effect by correcting for the GIs of nearby genes. However, we only corrected for GIs of genes within 1 Mb of the current index gene, leaving the possibility of pleiotropic effects beyond this threshold. For example, the GI of an index gene may influence both the expression of the index gene and another gene, located outside of the 1 Mb window, where the induced changes in that genes' expression are the causal factor of the identified target genes. A related problem arises when a shared genetic component between neighbouring index genes causes all of them to associate with a single distant target gene, hindering the identification of the index gene responsible for the induced *trans*-effect. By correcting for the GI of nearby genes, these potentially biologically relevant effects are lost (Fig. 1).

In conclusion, we present a genome-wide approach that identifies causal effects of gene expression on distal transcriptional activity in population genomics data and showcase several examples providing new biological insights. The resulting resource is available as an interactive network browser that can be utilized by researchers for look-ups of specific genes of interest (see URLs).

## Methods

**Cohorts.** The Biobank-based Integrative Omics Study (BIOS, Additional SI1) Consortium comprises six Dutch biobanks: Cohort on Diabetes and Athero-sclerosis Maastricht[64] (CODAM), LifeLines-DEEP[65] (LLD), Leiden Longevity Study[66] (LLS), Netherlands Twin Registry[67] (NTR), Rotterdam Study[68] (RS), Prospective ALS Study Netherlands[69] (PAN). The data that were analysed in this study came from 3072 unrelated individuals (Supplementary Data 1). Genotype data and gene expression data were measured in whole blood for all samples. In addition, sex, age, and cell counts were obtained from the contributing cohorts. The Human Genotyping facility (HugeF, Erasmus MC, Rotterdam, The Netherlands, http://www.blimdna.org) generated the RNA-sequencing data.

**Genotype data.** Genotype data were generated within each cohort (LLD: Tig-chelaar et al.[65]; LLS: Deelen et al.[70]; NTR: Lin et al.[71]; RS: Hofman et al.[68]; PAN: Huisman et al.[69]).

For each cohort, the genotype data were harmonized toward the Genome of the Netherlands[72] (GoNL) using Genotype Harmonizer[73] and subsequently imputed per cohort using Impute2[74] and the GoNL reference panel[72] (v5). We removed SNPs with an imputation info-score below 0.5, a HWE $P < 10^{-4}$, a call rate below 95% or a minor allele frequency smaller than 0.01. These imputation and filtering steps resulted in 7,545,443 SNPs that passed quality control in each of the datasets.

**Gene expression data.** Total RNA from whole blood was deprived of globin using Ambion's GLOBIN clear kit and subsequently processed for sequencing using Illumina's Truseq version 2 library preparation kit. Paired-end sequencing of 2 × 50 bp was performed using Illumina's Hiseq2000, pooling 10 samples per lane. Finally, read sets per sample were generated using CASAVA, retaining only reads passing Illumina's Chastity Filter for further processing. Data were generated by the Human Genotyping facility (HugeF) of ErasmusMC (The Netherlands, see URLs). Initial QC was performed using FastQC (v0.10.1), removal of adaptors was performed using cutadapt[75] (v1.1), and Sickle[76] (v1.2) was used to trim low quality ends of the reads (minimum length 25, minimum quality 20). The sequencing reads were mapped to human genome (HG19) using STAR[77] (v2.3.0e).

To avoid reference mapping bias, all GoNL SNPs (http://www.nlgenome.nl/?page_id=9) with MAF > 0.01 in the reference genome were masked with N. Read pairs with at most 8 mismatches, mapping to as most 5 positions, were used.

Gene expression quantification was determined using base counts[17]. The gene definitions used for quantification were based on Ensembl version 71, with the extension that regions with overlapping exons were treated as separate genes and reads mapping within these overlapping parts did not count towards expression of the normal genes.

For data analysis, we used counts per million (CPM), and only used protein coding genes with sufficient expression levels (median log(CPM) > 0), resulting in a set of 10,781 genes. To limit the influence of any outliers still present in the data, the data were transformed using a rank-based inverse normal transformation within each cohort.

**Constructing a local genetic instrument for gene expression.** We started by constructing genetic instruments (GIs) for the expression of each gene in our data. We first split up the genotype and RNA-sequencing data in a training set (one-third of all samples, $N_{\text{train}} = 1021$) and a test set (two-thirds of all samples, $N_{\text{test}} = 2051$), making sure all cohorts and both sexes were evenly distributed over the train and test sets (57% female), as well as an even distribution of age (mean = 56, sd = 14.8). Using the training set only, we built a GI for each gene $j$ separately that best predicts its expression levels using lasso, using nearby genetic variants only (either within the gene or within 100 kb of a gene's TSS or TES), while correcting for both known (cohort, sex, age, cell counts) and unknown covariates:

$$y_j = \boldsymbol{D}^T\beta + \boldsymbol{C}^T\gamma + \epsilon \tag{1}$$

where $y_j$ is the gene expression for gene $j$, $\boldsymbol{D}$ the scaled matrix with dosage values for the nearby genetic variants with its corresponding regression coefficients $\beta$, $\boldsymbol{C}$ the matrix of scaled known and unknown covariates and their corresponding regression coefficients $\gamma$, and the vector or residuals $\epsilon$. Estimation of the unknown covariates was done by applying cate[14] to the observed expression data, leading to five unknown latent factors used. Those factors, together with the known covariates, were left unpenalized. To estimate the optimal penalization parameter $\lambda$, we used five-fold cross-validation as implemented in the R package glmnet[78]. The obtained GI for index gene $j$ consisted of a weighted linear combination of the dosage values of the selected nearby genetic variants, weighted by the obtained regression coefficients $\beta$, to obtain $GI_j$ for index gene $j$:

$$GI_j = \boldsymbol{D}^T\beta \tag{2}$$

where $GI_j$ is a vector of values. We then evaluated its predictive ability in the test set by employing Analysis of Variance (ANOVA) to evaluate the added predictive power of the GI over the covariates and neighbouring GIs (within 1 Mb), as reflected by the $F$-statistic ($F > 10$).

Earlier work related to establishing putative causal relations between gene expression and phenotypic traits[7,8] shows overlap with our proposed method, but

also some distinct differences. First, none of them attempt to account for pleiotropy. Furthermore, two earlier studies[7,8] have both used a single top eQTL SNP as a GI, or have used all nearby genetic variants, without feature selection[8]. While not performing feature selection at all may improve the predictive ability over our method, it may also induce pleiotropy or LD. This may especially be the case since the authors have used a 1 Mb window around a gene, and have not corrected for pleiotropy or LD. The other study[7] has indeed used feature selection using elastic net, which also leads to sparse models, albeit slightly less sparse than our proposed method.

**Testing for *trans*-effects.** Using linear regression, we tested for an association between each GI $j$ and the expression of potential target genes $k$ *in trans* (>10 Mb), while correcting for known (cohort, sex, age, red, and white blood cell counts) and unknown covariates, as well as GIs of nearby genes (<1 Mb):

$$y_k = GI_j\varphi_j + \boldsymbol{C}^T\gamma + \boldsymbol{G}_j^T + \epsilon \tag{3}$$

where we test for the significance of the regression coefficient $\varphi_j$, and $\boldsymbol{G}_j$ represents the GIs of index genes near the current index gene $j$. Missing observations in the measured red blood cell count (RBC) and white blood cell counts (WBC) were imputed using the R package pls, as described earlier[5]. Any inflation or bias in the test-statistics was estimated and corrected for using the R package bacon[5]. Correction for multiple testing was done using Bonferroni (Wald $P < 7 \times 10^{-10}$). The resulting networks were visualized using the R packages network and ndtv.

**Enrichment analyses.** Functional analysis of gene sets was performed for GO Molecular Function annotations using DAVID[79], providing a custom background consisting of all genes with a predictive GI ($F > 10$). Fisher's exact test was employed to specifically test for an enrichment of transcription factors using manually curated database of transcription factors[22].

**Simulation study.** Simulating data of genetic instruments (GIs), their corresponding gene expression measurements, and a target gene was done as follows:

- Generate two normally distributed, correlated genetic instruments, where the correlation between the different GIs represents LD/pleiotropy. We used five different values for the correlation $r_{\text{GI}}$ as estimated in our data, corresponding to the minimum absolute correlation in our identified effects, the 25th, 50th, 75th percentile, and the maximum value.
- Generate the index gene expression by creating a new normally distributed variable correlated to the index GI. Again, we used 5 different values for the correlation $r_{\text{GI, index}}$, using estimations from our data, corresponding to the minimum absolute correlation in our identified effects, the 25th, 50th, 75th percentile, and the maximum value.
- Similarly, generate the nearby gene expression by creating a new normally distributed variable correlated to the nearby GI. Here, we also used 5 values for the correlation $r_{\text{GI, nearby}}$ corresponding to the minimum absolute correlation in our identified effects, the 25th, 50th, 75th percentile, and the maximum value.
- Lastly, generate the target gene by creating a new normally distributed variable correlated to either the index gene ($r_{\text{index, target}}$), or the nearby gene ($r_{\text{nearby, target}}$), depending on the hypothesis tested (Supplementary Fig. 6). We again used different values for these correlations.

We have simulated two scenarios (see Supplementary Fig. 6), corresponding to the alternative and null hypotheses:

- The GI of the index gene causally influences its corresponding index gene, which influences the target gene (Supplementary Fig. 6a).
- The GI of a nearby gene causally influences its corresponding gene, which influences the target gene (Supplementary Fig. 6b).

For both scenarios, we have tested the effect of the index GI ($\beta_{\text{index}}$) on the target gene $y$, both corrected for LD by including the GI of the nearby gene $GI_{\text{nearby}}$,

$$y = \beta_{\text{index}}GI_{\text{index}} + GI_{\text{nearby}} + \varepsilon \tag{4}$$

and without correcting for LD.

$$y = \beta_{\text{index}}GI_{\text{index}} + \varepsilon \tag{5}$$

For each set of different settings (i.e., different correlations among the different variables), this lead to the testing of four models, two for each scenario (Supplementary Fig. 6). Repeating this analysis 500 times for each unique set of settings, we then were able to estimate the power of each model by calculating the proportion of times the $P$-value was smaller than 0.05:

$$\text{power} = 1/500 \sum_{i=1}^{500} I(P_i < 0.05) \tag{6}$$

**URLs**. Look-ups can be performed using our interactive gene network browser at http://bbmri.researchlumc.nl/NetworkBrowser/. Data were generated by the Human Genotyping facility (HugeF) of ErasmusMC, the Netherlands (http://www.glimDNA.org). Webpages of participating cohorts: LifeLines, http://lifelines.nl/lifelines-research/general; Leiden Longevity Study, http://www.healthy-ageing.nl/ and http://www.leidenlangleven.nl/; Netherlands Twin Registry, http://www.tweelingenregister.org/; Rotterdam Studies, http://www.erasmusmc.nl/epi/research/The-Rotterdam-Study/; Genetic Research in Isolated Populations program, http://www.epib.nl/research/geneticepi/research.html#gip; CODAM study, http://www.carimmaastricht.nl/; PAN study, http://www.alsonderzoek.nl/.

**Code availability**. R code is available from https://git.lumc.nl/r.luijk/DirectedGeneNetworks. This repository describes the main analyses done.

**Data availability**. Raw data were submitted to the European Genome-phenome Archive (EGA) under accession EGAS00001001077.

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

## Acknowledgements

This research was financially supported by BBMRI-NL, a Research Infrastructure financed by the Dutch government (NWO, numbers 184.021.007 and 184.033.111). Samples were contributed by LifeLines, the Leiden Longevity Study, the Netherlands Twin Registry (NTR), the Rotterdam Study, the Genetic Research in Isolated Populations program, the Cohort on Diabetes and Atherosclerosis Maastricht (CODAM) study and the Prospective ALS study Netherlands (PAN). We thank the participants of all aforementioned biobanks and acknowledge the contributions of the investigators to this study. This work was carried out on the Dutch national e-infrastructure with the support of SURF Cooperative. We acknowledge the support from the Netherlands CardioVascular Research Initiative (the Dutch Heart Foundation, Dutch Federation of University Medical Centres, the Netherlands Organisation for Health Research and Development, and the Royal Netherlands Academy of Sciences) for the GENIUS project Generating the best evidence-based pharmaceutical targets for atherosclerosis (CVON2011-19).

## Author contributions

Conceptualization, B.T.H., E.W.v.Z., R.L., K.F.D., M.v.I.; Methodology, R.L., W.E.v.Z., M. v.I.; Formal Analysis, R.L.; Resources, W.A., A.C., D.I.B., C.M.v.D., M.M.J.v.G., J.H.V., C. W., L.F., P.A.C.t.H., R.J., J.v.M., H.M., P.E.S., BIOS Consortium; Writing—Original Draft, R.L.; Writing—Review and Editing, R.L., B.T.H., E.W.v.Z., P.H., A.C., D.I.B., C.M. v.D., M.M.J.v.G., J.H.V., C.W., P.A.C.t.H., R.J., J.v.M., H.M., P.E.S.; Visualization, R.L., B. T.H.; Supervision, B.T.H., E.W.v.Z.

## Additional information

**Competing interests:** The authors declare no competing interests.

**BIOS (Biobank-based Integrative Omics Study) Consortium**

Marian Beekman[1], Ruud van der Breggen[1], Joris Deelen[1], Nico Lakenberg[1], Matthijs Moed[1], H. Eka D. Suchiman[1], Wibowo Arindrarto[2], Peter van 't Hof[2], Marc Jan Bonder[3], Patrick Deelen[3], Ettje F. Tigchelaar[3], Alexandra Zhernakova[3], Dasha V. Zhernakova[3], Jenny van Dongen[4], Jouke J. Hottenga[4], René Pool[4], Aaron Isaacs[5], Bert A. Hofman[5], Mila Jhamai[6], Carla J.H. van der Kallen[7], Casper G. Schalkwijk[7], Coen D.A. Stehouwer[7], Leonard H. van den Berg[8], Michiel van Galen[9], Martijn Vermaat[9], Jeroen van Rooij[11], André G. Uitterlinden[11], Michael Verbiest[11], Marijn Verkerk[11], P. Szymon M. Kielbasa[12], Jan Bot[13], Irene Nooren[13], Freerk van Dijk[14], Morris A. Swertz[14] & Diana van Heemst[15]

[13]SURFsara, Amsterdam 1098 XG, The Netherlands. [14]Genomics Coordination Center, University Medical Center Groningen, University of Groningen, Groningen 9713 AV, The Netherlands. [15]Department of Gerontology and Geriatrics, Leiden University Medical Center, Leiden 2333 ZC, The Netherlands

