## [Peer Review File · Nature Communications]

Reviewers' comments:

Reviewer #1 (Remarks to the Author):

This is very interesting paper.

I find the idea of using MR ideas to find multi-SNP anchor GIs to be compelling. The "checks" are better described as sensitivity analyses.

The major issue I see with the paper is lack of detail in the methods. I would like to see the following added.

1. The mathematical form of the analyses, and the parameters, need to formally stated.
2. Comparison between the multi-SNPs models and the various gene expression imputation methods (PREDIXCAN, TWAS, etc.) multi-SNP models is also needed. A specific section on this should be inserted in supplementary methods or in main methods.
3. A supplementary methods file giving more details of the methods should be added.

Reviewer #2 (Remarks to the Author):

This manuscript presents a mendelian randomisation analysis of gene expression data, using cis-eQTL as genetic instrument variables to link the expression of a given gene to the expression of another gene located elsewhere in the genome. The idea is that a directed graph can be constructed among genes, leading to the identification of genes that act as promoters for the expression of other genes. A mediation analysis is then also conducted, where it is assumed that methylation mediates gene expression. I find the manuscript to be interesting and well written, but I have a number of questions and some suggestions for improvement.

1. Testing of "statistical causality" occurs in both the testing and training set but should only occur in the testing set. The analysis of 6,600 GIs with 10,781 genes in trans was conducted on all 3,072 samples. It should only be done on the 2,050 testing individuals. This is because GIs identified above a statistical threshold are likely those where the true effect and the error align. By testing for causality in all of the data those errors are carried through so any covariance between estimation error and expression in trans may influence the analysis. Keeping the testing and training sets separate would eliminate this potential source of bias. This bias is represented both in the GIs identified are a biased subset and their effects are biased.
2. I think it would be interesting to test bi-directionality for the interactions identified. This could identify feedbacks in the system and help strengthen the results.
3. More information is needed on the mediation analysis. My concern here again is that the data used to determine the GI for methylation and the data used to determine the mediation on expression are the same. Also the analysis seems disjoint and tacked onto the paper. None of the results are presented in the main manuscript. Also, why not test mediation of expression on methylation - some sites are methylated throughout life (i.e. in response to the environment, or with age)?
4. Generally more methodological details are needed within the methods section. Is the bonferroni correction taking into account off of the tests conducted in the paper? Please be clear in each section of the methods the exact sample sizes used.

5. Rather than testing whether the GI's are associated with cell counts, could cell count not be adjusted from the expression and methylation data prior to analysis? Would this not remove this variation potentially leading to the identification of more signals.

Reviewer #3 (Remarks to the Author):

In the manuscript, Luijk et al. use genetic instruments to identify index genes that are likely a mediators of target genes in trans. They analyze gene expression data of ~3K samples to identify 49 genes that affect gene expression in trans, and build directional gene networks, while counting for confounding, linkage disequilibrium, and pleiotropy. Overall, I find the method and analyses described in this manuscript sound, and the result insightful. The use of a web browser to present the results will be extremely helpful to the community. The manuscript would benefit from an unambiguous definition of terms, a clearer description of methods, and a discussion of a recently proposed method GMAC (Yang et al., Genome Research 2017).

Major comments

1) The term "genetic instrument" (GI) seems to have different meanings in different parts of the manuscript. For example, in Introduction (line 81 to 85), GI seems to be defined as the local genetic component of gene expression (i.e. predicted cis regulated gene expression). However, in the legend of Figure 1 (line 793 to 794), GI is defined as "nearby SNPs with a possible effect on the nearby gene". And in the method section (line 481 to 483), GI is defined as the weighted combination of genotype dosages. Since the manuscript heavily relies on the concept of GI, I think it's crucial that the authors give an unambiguous mathematical definition of genetic instruments in the context of this manuscript. Throughout the manuscript, I presume that GI is the predicted local genetic component of gene expression.

2) In the section "Constructing a local genetic instrumental variable for gene expression", may I confirm that the authors learn the LASSO weights using one-third of the samples, and perform association between GI and trans gene expression **only** on the remaining two-thirds of the samples? In other words, is the one-third of the samples **only** used for obtaining weights, and **not** for testing association between GI and trans expression? I am concerned that if the one-third samples used for learning LASSO weights are also used for association testing, correlation between environmental and confounding effects from these samples can cause false associations (since the weights are affected by environmental and confounding effects, which can seep into GI). Alternatively, the authors may benefit from weights (e.g. Gusev et al. Nat Genet 2016) obtained from an independent dataset. Using weights from an external data set is akin to performing a 2-sample Mendelian randomization, and could remove false associations caused by obtaining weights and testing for associations on the same set of samples.

3) The above comment also applies to testing the mediation effect of DNA methylation on target gene expression. The author would benefit from estimating weights and testing for associations on different set of samples.

4) Recently, Yang et al. published a paper titled "Identifying cis-mediators for trans-eQTLs across many human tissues using genomic mediation analysis" in Genome Research. In this manuscript, the authors propose the GMAC algorithm, which applies rigorous test to filter out potential confounders and uses single eQTLs as instrumental variables to identify mediation effects, and identify substantial amount of mediations across 44 tissues. I am a bit surprised that Yang et al.'s paper isn't mentioned at all in the manuscript. The authors may benefit from a comparison with Yang et al's paper both in terms of methods and results. If the two methods are not comparable,

the authors may still benefit from a brief discussion of previous findings.

5) Although the authors apply various safety checks to ensure the validity of the results, I highly encourage the authors to perform simulations to demonstrate that their approach is indeed well-calibrated under the null, has high power to detect associations, and controls for LD and pleiotropic effect. A thorough simulation can add more credibility to the results obtained from real-data analysis.

Minor comments

1) Throughout the manuscript, the letter "N" has been used to represent both sample size, number of genes, number of GI's. This can be a bit confusing and difficult to follow some times. I suggest the authors to be explicit with N. For example, when referring to sample size, use "N_sample", and when referring to number of genes, use "N_gene" instead.

2) Given the fact that causality is in general very difficult to prove, I suggest replace "causality" with "putative causality" following Pickrell et al Nat Genet 2016.

3) To better present the method, e.g. what goes into the LASSO model, and what covariates are included in association testing, I suggest the authors to use mathematical equations. These will make the method section much easier to follow and understand.

Response to Reviewers' comments:

Reviewer #1 (Remarks to the Author):

This is very interesting paper.

I find the idea of using MR ideas to find multi-SNP anchor GIs to be compelling. The "checks" are better described as sensitivity analyses.

The major issue I see with the paper is lack of detail in the methods. I would like to see the following added.

1. The mathematical form of the analyses, and the parameters, need to formally stated.

In order to clearly explain all performed analyses, we have significantly expanded upon the Methods section, including mathematical formulations.

2. Comparison between the multi-SNPs models and the various gene expression imputation methods (PREDIXCAN, TWAS, etc.) multi-SNP models is also needed. A specific section on this should be inserted in supplementary methods or in main methods.

We have included a paragraph in the Methods section detailing the differences and overlapping features of our proposed method and those proposed earlier (Gusev et al., Gamazon et al.)

3. A supplementary methods file giving more details of the methods should be added.

Given the improvements made to the Methods section, we believe this now clearly explains the methodology used, and hence does not necessitate a supplementary methods section.

Reviewer #2 (Remarks to the Author):

This manuscript presents a mendelian randomisation analysis of gene expression data, using cis-eQTL as genetic instrument variables to link the expression of a given gene to the expression of another gene located elsewhere in the genome. The idea is that a directed graph can be constructed among genes, leading to the identification of genes that act as promoters for the expression of other genes. A mediation analysis is then also conducted, where it is assumed that methylation mediates gene expression. I find the manuscript to be interesting and well written, but I have a number of questions and some suggestions for improvement.

1. Testing of "statistical causality" occurs in both the testing and training set but should only occur in the testing set. The analysis of 6,600 GIs with 10,781 genes in trans was conducted on all 3,072 samples. It should only be done on the 2,050 testing individuals. This is because GIs identified above a statistical threshold are likely those where the true effect and the error align. By testing for causality in all of the data those errors are carried through so any covariance between estimation error and expression in trans may influence the analysis. Keeping the testing and training sets separate would eliminate this potential source of bias. This bias is represented both in the GIs identified are a biased subset and their effects are biased.

We have re-analyzed the data using the 2,051 samples from the test set only, showing very similar results to the original analysis using the full set of 3,071 samples. The figure below shows how the only difference in the analyses is a limited statistical power in the test set. We now discuss this analysis in the text, and the figure below is added as a supplementary figure in the manuscript.

2. I think it would be interesting to test bi-directionality for the interactions identified. This could identify feedbacks in the system and help strengthen the results.

In the analysis, we test all possible *trans*-interactions between the 6,600 predictive genetic instruments (GIs with $F > 10$) and the 10,781 genes. As such, if there is a possible bi-directional effect, it is already tested in the form of two tests for unidirectional effects. However, no such bi-directional effects were identified.

3. More information is needed on the mediation analysis. My concern here again is that the data used to determine the GI for methylation and the data used to determine the mediation on expression are the same. Also the analysis seems disjoint and tacked onto the paper. None of the results are presented in the main manuscript. Also, why not test mediation of expression on methylation - some sites are methylated throughout life (i.e. in response to the environment, or with age)?

Upon re-reading, we agree with the reviewer that the mediation analysis seems disjoint. In compliance with the reviewer's criticism, we removed the section altogether, since it does not add to the main message of the paper. The other comment on the samples used for the different stages of the analysis are still valid. However, similar to the answer on comment 1, we find that testing the mediation in the test set only attenuates the test statistics, in line with a reduction in sample size (see figure below), while leaving the overall conclusion the same.

4. Generally more methodological details are needed within the methods section. Is the bonferroni correction taking into account off of the tests conducted in the paper? Please be clear in each section of the methods the exact sample sizes used.

In order to clearly explain all performed analyses, we have significantly improved upon the Methods section by explaining the analyses in more detail: this includes adding mathematical formulations where appropriate, explicitly specifying sample sizes.

5. Rather than testing whether the GI's are associated with cell counts, could cell count not be adjusted from the expression and methylation data prior to analysis? Would this not remove this variation potentially leading to the identification of more signals.

In our original analysis, we have already corrected for cell counts. However, we have made this clearer in the manuscript text to avoid any confusion.

Reviewer #3 (Remarks to the Author):

In the manuscript, Luijk et al. use genetic instruments to identify index genes that are likely a mediators of target genes in trans. They analyze gene expression data of ~3K samples to identify 49 genes that affect gene expression in trans, and build directional gene networks, while counting for confounding, linkage disequilibrium, and pleiotropy. Overall, I find the method and analyses described in this manuscript sound, and the result insightful. The use of a web browser to present the results will be extremely helpful to the community. The manuscript would benefit from an unambiguous definition of terms, a clearer description of methods, and a discussion of a recently proposed method GMAC (Yang et al., Genome Research 2017).

Major comments

1) The term “genetic instrument” (GI) seems to have different meanings in different parts of the manuscript. For example, in Introduction (line 81 to 85), GI seems to be defined as the local genetic component of gene expression (i.e. predicted cis regulated gene expression). However, in the legend of Figure 1 (line 793 to 794), GI is defined as “nearby SNPs with a possible effect on the nearby gene”. And in the method section (line 481 to 483), GI is defined as the weighted combination of genotype dosages. Since the manuscript heavily relies on the concept of GI, I think it’s crucial that the authors give an unambiguous mathematical definition of genetic instruments in the context of this manuscript. Throughout the manuscript, I presume that GI is the predicted local genetic component of gene expression.

We now give mathematical formulations of key concepts in the Methods section, making it easier to understand the methodology used. In addition, we have edited the main text, uniformly defining the term “genetic instrument” as to avoid any confusion about its meaning.

2) In the section “Constructing a local genetic instrumental variable for gene expression”, may I confirm that the authors learn the LASSO weights using one-third of the samples, and perform association between GI and trans gene expression ****only**** on the remaining two-thirds of the samples? In other words, is the one-third of the samples ****only**** used for obtaining weights, and ****not**** for testing association between GI and trans expression? I am concerned that if the one-third samples used for learning LASSO weights are also used for association testing, correlation between environmental and confounding effects from these samples can cause false associations (since the weights are affected by environmental and confounding effects, which can seep into GI). Alternatively, the authors may benefit from weights (e.g. Gusev et al. Nat Genet 2016) obtained from an independent dataset. Using weights from an external data set is akin to performing a 2-sample Mendelian randomization, and could remove false associations caused by obtaining weights and testing for associations on the same set of samples.

We have re-analyzed the data using the 2,051 samples from the test set only, showing very similar results to the original analysis using the full set of 3,071 samples. The figure below shows how the only difference in the analyses is a limited statistical power in the test set. We now discuss this analysis in the text, and the figure below is added as a supplementary figure in the manuscript.

3) The above comment also applies to testing the mediation effect of DNA methylation on target gene expression. The author would benefit from estimating weights and testing for associations on different set of samples.

After having considered this, and another reviewer's similar comment, we removed the section on mediation altogether, as we feel it does not add to the main message of the manuscript. The above comment on the samples used for the different stages of the analysis are still valid. However, similar to the answer on comment 2, we find that testing the mediation in the test set only attenuates the test statistics, while leaving the overall conclusion the same.

4) Recently, Yang et al. published a paper titled “Identifying cis-mediators for trans-eQTLs across many human tissues using genomic mediation analysis” in Genome Research. In this manuscript, the authors propose the GMAC algorithm, which applies rigorous test to filter out potential confounders and uses single eQTLs as instrumental variables to identify mediation effects, and identify substantial amount of mediations across 44 tissues. I am a bit surprised that Yang et al.’s paper isn’t mentioned at all in the manuscript. The authors may benefit from a comparison with Yang et al.’s paper both in terms of methods and results. If the two methods are not comparable, the authors may still benefit from a brief discussion of previous findings.

In our Discussion section, we compare our methods to several other efforts using cis-mediated trans-eQTLs, where we have now included the suggested paper by Yang et al. In essence, these methods are quite distinct. Our method explicitly aims at estimating genetic instruments for index genes (cis-genes). Next, we mention how these methods often suffer from limited statistical power, and how they do not take into account LD and pleiotropy, leading to the identification of several cis-genes seemingly mediating a trans-eQTL.

5) Although the authors apply various safety checks to ensure the validity of the results, I highly encourage the authors to perform simulations to demonstrate that their approach is indeed well-

calibrated under the null, has high power to detect associations, and controls for LD and pleiotropic effect. A thorough simulation can add more credibility to the results obtained from real-data analysis.

We have performed simulations to evaluate the power and effectiveness of correcting for LD/pleiotropy. Simulating data of genetic instruments (GIs), their corresponding gene expression measurements, and a target gene was done as follows:

- Generate two normally distributed, correlated genetic instruments. The correlation between the different GIs is going to simulate LD/pleiotropy. Here, we used 5 different values for the correlation r_{GI} corresponding to the minimum absolute correlation in our identified effects, the 25th, 50th, 75th percentile, and the maximum value.
- Generate the index gene expression by creating a new normally distributed variable correlated to the index GI. Here, we used 5 different values for the correlation $r_{GI, index}$, corresponding to the minimum absolute correlation in our identified effects, the 25th, 50th, 75th percentile, and the maximum value.
- Similarly, generate the nearby gene expression by creating a new normally distributed variable correlated to the nearby GI. Here, we also used 5 values for the correlation $r_{GI, nearby}$ corresponding to the minimum absolute correlation in our identified effects, the 25th, 50th, 75th percentile, and the maximum value.
- Lastly, generate the target gene by creating a new normally distributed variable correlated to either the index gene ($r_{index, target}$), or the nearby gene ($r_{nearby, target}$), depending on the hypothesis tested. We again used different values for these correlations.

We have simulated two scenarios (see figure below), corresponding to the alternative and null hypotheses:

- A. The GI of the *index gene* causally influences its corresponding index gene, which influences the target gene.
- B. The GI of a *nearby gene* causally influences its corresponding gene, which influences the target gene.

For both scenarios we have tested the effect of the index GI (β_{index}) on the target gene y , both corrected for LD by including the GI of the nearby gene GI_{nearby} ,

$$y = \beta_{index} * GI_{index} + GI_{nearby} + \varepsilon$$

and without correcting for LD.

$$y = \beta_{index} * GI_{index} + \varepsilon$$

For each set of different settings (i.e. different correlations among the different variables), this led to the testing of four models, two for each scenario A and B. Repeating this analysis 500 times for each unique set of settings, we then were able to estimate the power of each model by calculating the proportion of times the P-value was smaller than 0.05:

$$power = 1/500 \sum_{i=1}^{500} I(P_i < 0.05)$$

The figure below shows the statistical power for different values for the different correlations r_{GI} , $r_{GI, index}$, $r_{GI, nearby}$, $r_{index, target}$. The red and green lines show the results under the alternative hypothesis (uncorrected for LD, and corrected for LD, respectively), i.e. when the GI of the index gene causally affects the target gene. The blue and purple show the results under the null hypothesis.

Overall, the analysis shows high power to detect an effect of the GI of the index gene on the target gene. As expected, the correlation between GI and index gene, and between index gene and target gene contribute most to an increased power. The presence of correlated GIs of nearby genes plays a smaller role.

Under the null hypothesis (i.e. when a neighboring gene influences the target gene, and not the tested index gene, see blue and purple lines), the uncorrected analysis will indeed lead to false positives (indicated by higher power), while the corrected analysis will indeed lead to false positives in 5% of the tests performed, as expected.

While we agree that a simulation analysis may provide more insight into how our method is expected to perform under different circumstances, we do feel as though the main message of the paper should be about the principles of the methodology, and that this analysis does not fit well with that message. In addition, within the field it is known that Mendelian Randomization-type approaches need large sample sizes to detect effects. For these reasons, we have decided not to include these simulations in the manuscript at this time.

Minor comments

1) Throughout the manuscript, the letter “N” has been used to represent both sample size, number of genes, number of GI’s. This can be a bit confusing and difficult to follow some times. I suggest the authors to be explicit with N. For example, when referring to sample size, use “N_sample”, and when referring to number of genes, use “N_gene” instead.

Where possible, we have either changed “N” to suggested notations, or used different wording altogether.

2) Given the fact that causality is in general very difficult to prove, I suggest replace “causality” with “putative causality” following Pickrell et al Nat Genet 2016.

Following the reviewer’s suggestion, we have now more carefully described our aim to establish “causality” to “putative causality”.

3) To better present the method, e.g. what goes into the LASSO model, and what covariates are included in association testing, I suggest the authors to use mathematical equations. These will make the method section much easier to follow and understand.

In order to clearly explain all performed analyses, we have significantly improved upon the Methods section by including mathematical formulations.

Reviewers' comments:

Reviewer #1 (Remarks to the Author):

My apologies but the response about the comparability of your method with previous methods such as predixcan or TWAS is not sufficient. Verbally expressing the differences is not sufficient.

I want to see a comparison between your method and those - running those methods on your data. using their models in trans also. and demonstrating the problems they may have and why your method may be noteworthy and useful. Do the same things with these methods.

Reviewer #2 (Remarks to the Author):

I find the authors have improved the methods section substantially and it is now much clearer. I have two remaining suggestions:

1. I think the simulation study suggested by another reviewer should be placed in the Supplementary Information of the manuscript. I agree with the other reviewer that a properly conducted simulation study is very important to demonstrate the authors claims that their approach approves over other methods At present their manuscript contains bold claims about how their method improves previous efforts, but there is really no evidence to back up these claims. I am surprised at the reluctance of the authors to fully revise their work and I suggest that they include a simulation study that properly demonstrates that their methodology does what they claim and improves upon previous methodology - they claim their study is not methods focused, but in reality it is the novelty of their approach that makes the paper interesting.

2. Despite all reviewers requesting a comparison of different approaches in the empirical analysis the authors did not do this. Again I am surprised at their reluctance to properly explore the data and their approach.

3. As I previously suggested, I think a paper-wide significance threshold is most appropriate. The authors first conduct 8976 lasso models, the then test associations between 6600 GIs and 10781 genes (just over 71 million tests), they then do further tests after this. And yet despite all of this testing they consistently change the significance threshold throughout the paper. I strongly disagree with this approach. The authors cannot ignore the statistical testing they have done before when moving onto another analysis.

Reviewer #3 (Remarks to the Author):

I really appreciate the authors for putting in the effort to perform simulations.

All of my comments have been addressed in a satisfactory manner, and I believe the paper is ready for publication.

Response to Reviewers' comments:

First of all, we thank all reviewers for their time and effort. The previous comments by all three reviewers were key in improving the manuscript. Consequently, we genuinely regret we may have come across as unwilling or reluctant to explicitly compare the mentioned methods, this definitely never was our intention. We mistakenly assumed the reviewers asked for a verbal discussion of the methods, and hence limited our comparison to that discussion. We now empirically compare the predixcan and TWAS methods to our data (see different answers below and manuscript text), which we feel has strengthened the manuscript.

Reviewer #1 (Remarks to the Author):

My apologies but the response about the comparability of your method with previous methods such as predixcan or TWAS is not sufficient. Verbally expressing the differences is not sufficient.

We apologize; we had the – mistaken – impression that the reviewer was looking for a discussion of other methods. We are certainly not unwilling to compare those methods to ours, and we have now done so both empirically and in a simulation experiment.

I want to see a comparison between your method and those - running those methods on your data. using their models in trans also. and demonstrating the problems they may have and why your method may be noteworthy and useful. Do the same things with these methods.

We now include an empirical comparison of the different approaches, and have also added a similar scenario in the simulation study. Specifically, we include the methods proposed by Gusev et al (2016, Nat Genet) and Gamazon et al (2015, Nat Genet).

When creating the genetic instruments (GIs), the proposed methods differ in terms of their predictive power of the index gene, as indicated by the F-statistic in the test set (Figure S4). In particular, the predictive power of the GI on the index gene is often lower in the method proposed by Gusev et al than those proposed by Gamazon et al and ourselves.

Next, we compared the different approaches to testing trans-effects. Similar to our method, both Gamazon et al and Gusev et al propose a linear regression model, regressing target gene expression on the GI. However, neither the proposed models by Gamazon et al nor by Gusev et al correct for LD/pleiotropy by including the GIs of nearby index genes in the models. Testing for trans-effects using these models often identifies multiple neighboring GIs associated with a single target gene (Figure S5), even though they may not be causally related to the target gene. This observation is confirmed by our the simulation study (Figure S6).

Reviewer #2 (Remarks to the Author):

I find the authors have improved the methods section substantially and it is now much clearer. I have two remaining suggestions:

1. I think the simulation study suggested by another reviewer should be placed in the Supplementary Information of the manuscript. I agree with the other reviewer that a properly conducted simulation study is very important to demonstrate the authors claims that their approach improves over other methods. At present their manuscript contains bold claims about how their method improves previous efforts, but there is really no evidence to back up these claims. I am surprised at the reluctance of the authors to fully revise their work and I suggest that they include a simulation study that properly demonstrates that their methodology does what they claim and improves upon previous methodology - they claim their study is not methods focused, but in reality it is the novelty of their approach that makes the paper interesting.

Following the reviewer's suggestion, we now include the simulation study in the manuscript, adding a supplemental figure and detailing the simulations in a supplemental materials section. The simulation confirms that our method is more specific in identifying the causal gene than its competitors.

2. Despite all reviewers requesting a comparison of different approaches in the empirical analysis the authors did not do this. Again I am surprised at their reluctance to properly explore the data and their approach.

We now include an empirical comparison of the different approaches, and have also added a similar scenario in the simulation study. Specifically, we include the methods proposed by Gusev et al (2016, Nat Genet) and Gamazon et al (2015, Nat Genet).

When creating the genetic instruments (GIs), the proposed methods differ in terms of their predictive power of the index gene, as indicated by the F-statistic in the test set (Figure S4). Strikingly, the predictive power of the GI on the index gene is often lower in the methods proposed by Gusev et al than those proposed by Gamazon et al and our method.

Next, we compared the different approaches to testing trans-effects. Similar to our method, both Gamazon et al and Gusev et al propose a linear regression model, regressing target gene expression on the GI. However, neither the proposed models by Gamazon et al nor by Gusev et al correct for LD/pleiotropy by including the GIs of nearby index genes in the models. Testing for trans-effects using these models often identifies several neighboring GIs associated with a single target gene (Figure S5), even though they may not be causally related to the target gene. This observation is confirmed in the simulation study (Figure S6).

3. As I previously suggested, I think a paper-wide significance threshold is most appropriate. The authors first conduct 8976 lasso models, then test associations between 6600 GIs and 10781 genes (just over 71 million tests), they then do further tests after this. And yet despite all of this testing they consistently change the significance threshold throughout the paper. I strongly disagree with this approach. The authors cannot ignore the statistical testing they have done before when moving onto another analysis.

We agree that it is good practice to use a single paper-wide p-value, which we now set to $\sim 7 \times 10^{-10}$ (correcting for $6600 \times 10781 = 71$ million tests, as referred to by the reviewer). We now refrain from making any references to more lenient thresholds referring to a look-up p-value, which is based on 10781 tests ($P = 4.6 \times 10^{-6}$). This threshold corresponds to a researcher who would investigate the trans-effects of only a single gene, instead of all 6600 GIs, and look this gene up in our resource. While appropriate for a look-up, we agree that it is not relevant in the current manuscript.

Reviewer #3 (Remarks to the Author):

I really appreciate the authors for putting in the effort to perform simulations. All of my comments have been addressed in a satisfactory manner, and I believe the paper is ready for publication.

REVIEWERS' COMMENTS:

Reviewer #1 (Remarks to the Author):

Thanks for your revisions to the paper. I am now satisfied with the novelty of the method and the relevance of the results.

Reviewer #2 (Remarks to the Author):

The authors have addressed all of my concerns and I am happy to recommend publication.